# Low risk for diabetic complications in type 1 diabetes patients carrying a protective insulin gene variant

René van Tienhoven[1], Anh Nguyet Vu[1], John S. Kaddis[1], Bart O. Roep[2]*

**1** Department of Diabetes and Cancer Discovery Science, Arthur Riggs Diabetes & Metabolism Research Institute, Beckman Research Institute, City of Hope, Duarte, California, United States America, **2** Department of Internal Medicine, Leiden University Medical Center, Leiden, Netherlands

* b.o.roep@lumc.nl

## Abstract

Type 1 diabetes patients carrying a 'protective' insulin gene (*INS*) variant present a disease endotype with reduced insulin antibody titers, preserved beta cell function and improved glycemic control. We tested whether this protective *INS* variant associated with lowered risk for development of proliferative diabetic retinopathy (PDR) and diabetic kidney disease (DKD) as long-term diabetic complications. Insulin gene polymorphisms were evaluated in 1,363 type 1 diabetes patients participating in the Diabetes Control and Complications Trial/Epidemiology of Diabetes Interventions and Complications (DCCT/EDIC) study that compared intensive versus conventional insulin therapy in relation with development of PDR and DKD with a follow-up of over two decades. PDR and DKD were absent in type 1 diabetes patients carrying the protective *INS* variant and receiving intensive insulin therapy (the current standard of clinical care) 1–5 years from their diagnosis (n = 67; mean post-diagnosis follow up of 20.4 ± 1.6 years), versus 11 of 258 patients (4.3%) lacking this variant (20.4 ± 1.8 years follow up). In the secondary intervention group of the intensive therapy arm (1–15 years of disease), PDR was significantly less frequent in carriers of the protective INS variant than those without it (4 of 83 [4.8%] vs. 31 of 260 [11.9%]; p = 0.032; 26.1 ± 3.9 and 26.3 ± 4.1 years follow-up, respectively), whereas DKD frequencies were no different between those with or without this variant (5 of 83 [6.0%] vs. 11 of 260 [4.2%]). Carrying a copy of this protective *INS* variant further reduces the risk of diabetic complications achieved by intensive insulin therapy and marks a disease endotype with superior glycemic control, increased and extended beta cell function, and prevention of DKD and PDR.

## Introduction

There is a growing insight that type 1 diabetes patients and their disease differ [1], even despite receiving the current standard of care to achieve glycemic control. Therefore, a great, and largely unmet, need exists to identity markers and correlates of type 1 diabetes patient subpopulations with differential disease progression, preservation in beta cell function, and glycemic

and Phenotypes, under accession number
phs000086.v3.p1.

**Funding:** This study was supported by the Wanek
Family Project for Type 1 Diabetes (Director: BOR).

**Competing interests:** The authors have declared
that no competing interests exist.

control. This is essential for the design and allocation of personalized and precision medicine
strategies, including future immune intervention therapy and the prognosis of disease progression [2].

Gene variation may be used to serve such a purpose, as genetic risk scores have been developed underscoring this notion [3]. Genetic variation in the insulin gene (*INS*) is the second
largest contributor to genetic risk to type 1 diabetes (T1D) [4]. Differences have been identified
in a variable number of tandem repeats (VNTR) in the *INS* promotor region from which the
highest risk is associated with class I VNTR alleles, whereas class III VNTRs have been linked
to a dominantly protective effect. These *INS* polymorphisms are believed to influence the
expression of proinsulin, where increased thymic expression and decreased pancreatic expression associate with the protective class III VNTR haplotype; this contributes to improved central immune tolerance to proinsulin and protection from type 1 diabetes [5, 6]. This *INS*
promoter polymorphism is in strong linkage disequilibrium with single nucleotide polymorphisms (SNPs) at the 3' untranslated region (UTR) of the insulin gene [4], allowing to determine the *INS*-related risk by using tagging SNPs for the promoter polymorphisms and *vice
versa* [7–9].

Yet, *INS* associated protection is not complete and around 20% of type 1 diabetes patients
carries a copy of the 'protective' *INS* variant. Strikingly, pediatric patients developing type 1
diabetes, despite carrying this protective *INS* variant and receiving the current standard of clinical care, showed reduced insulin autoimmunity, preserved beta cell function, higher C-peptide levels, and improved glycemic control [9]. This points to the possibility that genetic
variation may contribute to diagnosis of disease variants and patient stratification.

It has been well established that intensive insulin treatment compared to conventional therapy can prevent and delay diabetes complications [10]. We hypothesized that type 1 diabetes
patients receiving this current standard of care and carrying a copy of the protective *INS* variant have a further reduced risk of developing diabetes complications compared to patients
only carrying the susceptible *INS* variant.

## Materials and methods

The association between *INS* variation and the probability of microvascular diabetes complications, proliferative diabetic retinopathy (PDR) and diabetic kidney disease (DKD) was investigated in participants of the Diabetes Control and Complications Trial and Epidemiology of
Diabetes Interventions and Complications (DCCT/EDIC) Study [10]. Following project
approval, DCCT/EDIC clinical and SNP study data on 1,441 individuals was downloaded
from the National Center for Biotechnology Information database of Genotypes and Phenotypes website, under accession number phs000086.v3.p1. The DCCT/EDIC trial contained
two treatment arms (intensive insulin therapy vs conventional therapy), and two cohorts (primary prevention and secondary intervention) within each arm (four groups in total). The
association between *INS* variation and complication probability was analyzed for each group
separately. The patients in the primary prevention cohort had T1D for 1–5 years without evidence of retinopathy and albumin secretion rates of <40mg/24h. The patients in the secondary intervention cohort had T1D for 1–15 years and at least one microaneurysm in either eye
and albumin secretion rates no greater than 200mg/24h. PDR was defined as any individual
with an Early Treatment Diabetic Retinopathy Study score of 12 or more or receiving scatter
laser treatment. DKD was defined as any individual with an estimated glomerular filtration
rate less than 60 mL/min/1.73m$^2$ on two or more consecutive visits, or an albumin excretion
rate greater than 300 mg/24h on one visit. The *INS* tagging SNP was determined by genotyping rs3842752 (also known as +1127PstI) in the 3' UTR of *INS*; the A allele represents the

**Table 1. DCCT/EDIC patient characteristics by insulin gene variant.**

| Characteristic | | DCCT/EDIC Patients by Insulin Variant | | | |
|---|---|---|---|---|---|
| | | Protective | | Susceptible | |
| | | n | Value* | n | Value* |
| Age at T1D Onset (years)† | | 275 | 20.8 (±8.0) | 1088 | 21.3 (±8.1) |
| Sex | Female | 133 | 48% | 509 | 47% |
| | Male | 142 | 52% | 579 | 53% |
| Race | White | 266 | 97% | 1047 | 96.4% |
| | Black | 7 | 2% | 22 | 2% |
| | Hispanic | 2 | 1% | 12 | 1% |
| | Asian/Pacific Islander | 0 | 0% | 6 | 0.5% |
| | American Indian/Alaska Native | 0 | 0% | 1 | 0.1% |
| BMI (kg/m²) | DCCT Start | 275 | 23.2 (±2.7) | 1088 | 23.5 (±2.8) |
| | DCCT End (Y9) | 253 | 25.4 (±3.4) | 1003 | 25.8 (±3.8) |
| | EDIC Start | 254 | 25.6 (±3.5) | 1002 | 25.8 (±3.7) |
| | EDIC Last Available (Y8) | 242 | 27.6 (±4.1) | 971 | 27.5 (±4.5) |
| HbA1c (%)†† | Intensive therapy arm | | | | |
| | DCCT Start | 150 | 8.9 (±1.7) | 518 | 8.9 (±1.5) |
| | DCCT End (Y9) | 149 | 7.5 (±1.1) | 515 | 7.4 (±1.1) |
| | Conventional treatment arm | | | | |
| | DCCT Start | 125 | 8.9 (±1.8) | 570 | 8.9 (±1.6) |
| | DCCT End (Y9) | 124 | 9.2 (±1.6) | 569 | 9.1 (±1.5) |
| Post disease diagnosis follow up period (years) | Intensive arm | | | | |
| | Primary prevention group | 67 | 20.4 (±1.6) | 258 | 20.4 (±1.8) |
| | Secondary intervention group | 83 | 26.1 (±3.9) | 260 | 26.3 (±4.1) |
| | Conventional treatment arm | | | | |
| | Primary prevention group | 66 | 20.2 (±2.0) | 292 | 19.8 (±2.7) |
| | Secondary intervention group | 59 | 25.6 (±4.6) | 278 | 25.1 (±4.5) |

* Mean ± SD reported for continuous variables; % reported for categorical variables

† Calculated as age minus T1D duration at baseline of DCCT

†† Listed in order from DCCT start to EDIC last available, mmol/mol values for those with the protective insulin variant are: A) 73.8, B) 67.2, C) 67.2, and D) 63.9. For those with the susceptible insulin variant, values were identical, *i.e.*: E) 73.8, F) 67.2, G) 67.2, and H) 63.9. Percentages were converted to mmol/mol using a DCCT to International Federation of Clinical Chemistry (IFCC) formula described elsewhere [11]. Reporting of standard deviation in mmol/mol is not possible because the formula only works for numbers between 4% and 13%.

protective and G the susceptible variant. SNP data was available for 1,363 individuals. Since our *a priori* predictions were directional (*i.e.*, protection reduces risk), statistical significance was determined by a one-sided log rank test with a significance level of p<0.05. Further details of the study cohort are provided in Table 1.

## Results

We report that patients carrying protective *INS* have a further reduced risk of developing diabetes complications compared to those lacking this gene variant. In the primary prevention group of the DCCT/EDIC study, none out of 67 intensively treated patients carrying the protective *INS* variant (homo- or heterozygous) developed PDR or DKD with follow-up of 20.4 ± 1.6 years after diagnosis. However, 11 out of 258 patients (4.3%) lacking the protective *INS* variant developed complications (PDR only n = 5, DKD only n = 3, or both PDR and

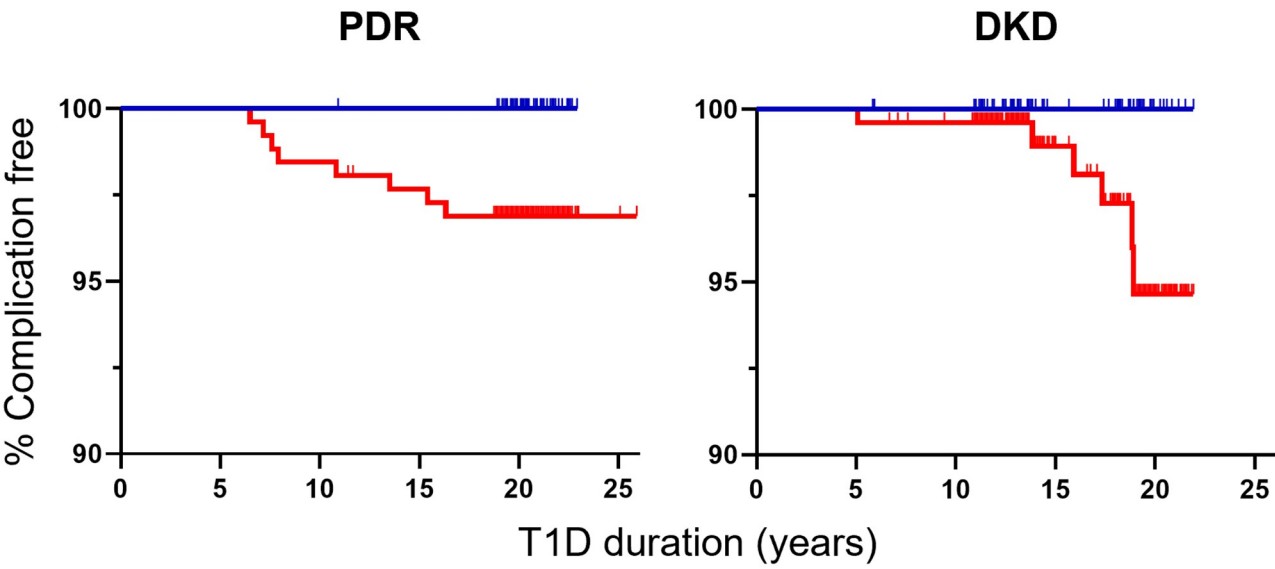

**Fig 1. Association between *INS* variation and complication probability during intensive insulin treatment.** The complication free probability of PDR (left) and DKD (right) is shown for type 1 diabetes patients carrying the protective (blue) and susceptible (red) *INS* variant in the primary prevention group of the DCCT/EDIC intensively treated study arm. None of the individuals carrying the protective *INS* variant developed PDR or DKD (0 of 67 and 0 of 67, respectively); however, PDR and DKD occurred in those carrying the susceptible *INS* variant (8 of 258 and 6 of 258, respectively). Homozygous and heterozygous protective patients were combined due to low numbers of protective homozygotes.

DKD n = 3) in 20.4 ± 1.8 years of follow-up after diagnosis (Fig 1). In the secondary intervention group, intensively treated patients with the protective *INS* variant showed significantly less PDR (4 of 83 [4.8%]; 26.1 ± 3.9 years post-diagnosis follow up) compared to patients with only the susceptible *INS* variant (31 of 260 [11.9%]; 26.3 ± 4.1 years post-diagnosis follow up; p = 0.032). However, there were no differences in the DKD frequencies between those with (5 of 83 [6.0%]) and without (11 of 260 [4.2%]) the protective INS variant (Fig 2). None of the seven patients homozygous for the protective *INS* variant developed DKD 22 ± 1 years after diagnosis. In 695 conventionally treated patients, complication frequencies did not differ between those with or without the protective *INS* variant (primary prevention group: 3 of 66 developed PDR (5%) vs. 19 of 292 (7%), respectively, and 4 of 66 developed DKD (6%) vs. 14 of 292 (5%), respectively; secondary intervention group: 13 of 59 developed PDR (22%) vs. 72 of 278 (26%), respectively, and 11 of 59 developed DKD (19%) vs. 34 of 278 (12%), respectively. Interestingly, none of the eight patients homozygous for the protective *INS* variant developed DKD up to 25.2 ± 4.9 years of post-diagnosis follow up.

## Discussion

We conclude that the protective *INS* variant further decreases the low risk for diabetes complications accomplished through intensive insulin treatment and glycemic control in patients with European ancestry. We propose that the combination of immune tolerance to insulin [5, 6] and preserved beta cell function [9] in patients carrying the protective *INS* variant contributes to superior glycemic control previously reported in carriers with recent onset pediatric T1D, thereby reducing the risk of development of diabetic complications. No complications were reported for patients carrying a protective *INS* variant in the primary prevention group. The lack of difference in complications frequencies between *INS* variants in the 'conventionally' treated group underscores that intensive treatment, which is the current standard of care, is necessary to reduce complications. The relationship between glycemic control and the

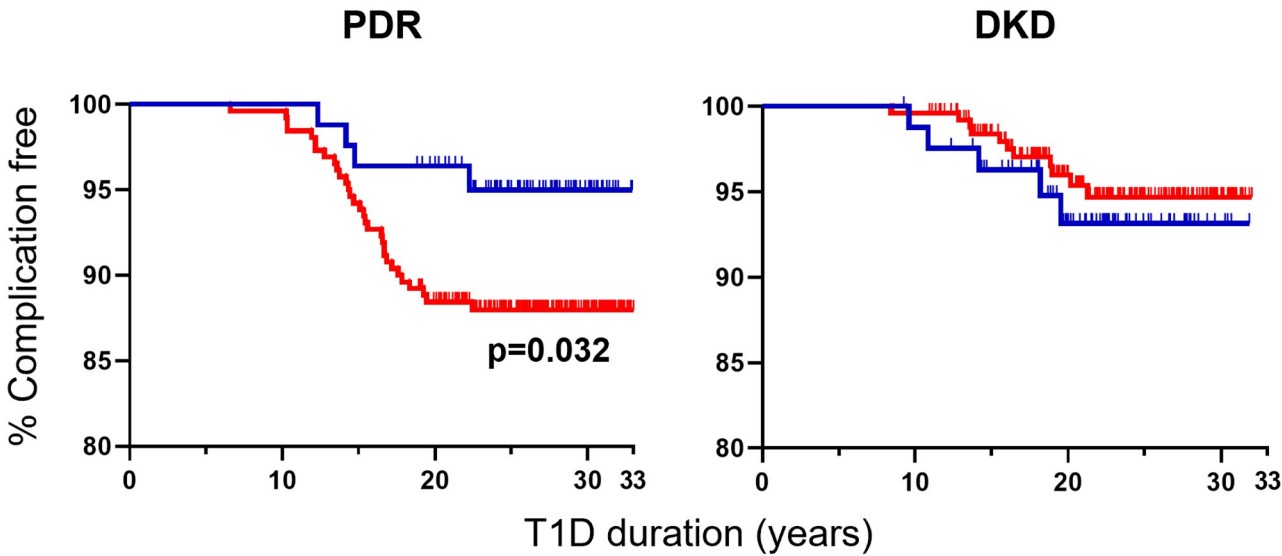

**Fig 2. Association between *INS* variation and complication probability during secondary intervention.** The complication free probability of PDR (left) and DKD (right) is shown for type 1 diabetes patients carrying the protective (blue) and susceptible (red) *INS* variant in the secondary intervention group of the DCCT/EDIC intensively treated study arm. Significantly fewer individuals carrying the protective *INS* variant developed PDR (4 of 83) compared to those with the susceptible *INS* variant only (31 of 260). The probability for developing DKD was not different between individuals carrying the protective *INS* variant (5 of 83) compared to those with the susceptible *INS* variant only (11 of 260). Homozygous and heterozygous protective patients were combined due to low numbers of protective homozygotes. Statistical significance was determined using a one-sided log rank test and indicated if p<0.05.

association between *INS* variation and diabetic complication remains unclear. In the DCCT cohort, HbA1c levels were much better in the intensive treatment arm compared to the conventional treatment arm but did not differ between patients with and without the protective *INS* variant within the intensive treatment arm. Yet, in the Hvidøre cohort an association between superior glycemic control and the protective *INS* variant was observed in children in the first year after their T1D diagnosis [9]. We speculate that the impact of the protective *INS* variant on complication probability involves epigenetic modifications early after disease onset ('metabolic memory') and may change with improved glycemic control later in disease progression [12]. Future and independent validation of our findings in other large cohorts therefore remains warranted.

We concede that the proportion of type 1 diabetes patients receiving the current standard of care ('intensive insulin therapy' in DCCT nomenclature) and nonetheless developing diabetic complications is thankfully relatively small. Yet, we argue that 'small numbers' are inherent to diagnosis of type 1 diabetes disease heterogeneity and personalized medicine, and relevant to that minor subset of type 1 diabetes patients involved. Although the DCCT/EDIC cohort is the largest T1D complications cohort with the longest follow-up time, a limitation of this study is the relatively small sample size. Alternative T1D cohorts were considered to replicate our results but these cohorts had even smaller numbers of study subjects, diabetic complications were not (yet) recorded or defined differently or the relevant SNP typing was missing. Fortunately, the Hvidøre pediatric cohort showed proof in the same direction that carrying a copy of the protective INS variant has favorable outcome: in their case on glycemic control and preserved beta cell function [9], both known to significantly reduce or even prevent diabetic complications, and thus indirectly yet independently supporting our observations in the DCCT cohort.

Our new data add to the favorable clinical prognosis in those carrying a copy of the 'protective' *INS* variant thereby linking this 'endotype' with superior glycemic control and reduced risk for diabetic complication. This finding may contribute to better selection and stratification of type 1 diabetes patients participating in immune intervention trials, and ultimately personalizing medicine.

## Acknowledgments

The authors thank Z. Chen and R. Natarajan for their advice in defining PDR and DKD.

## Author Contributions

**Conceptualization:** René van Tienhoven, John S. Kaddis, Bart O. Roep.

**Data curation:** René van Tienhoven, John S. Kaddis, Bart O. Roep.

**Formal analysis:** René van Tienhoven, Anh Nguyet Vu, John S. Kaddis, Bart O. Roep.

**Funding acquisition:** Bart O. Roep.

**Investigation:** René van Tienhoven, Bart O. Roep.

**Supervision:** Bart O. Roep.

**Validation:** René van Tienhoven.

**Visualization:** René van Tienhoven, Bart O. Roep.

**Writing – original draft:** René van Tienhoven, Anh Nguyet Vu, John S. Kaddis, Bart O. Roep.

**Writing – review & editing:** René van Tienhoven, John S. Kaddis, Bart O. Roep.

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
