## [Decision Letter · Decision Letter 0]

10 Nov 2022

PONE-D-22-27736Low risk for diabetic complications in type 1 diabetes patients carrying a protective insulin gene variantPLOS ONE

Dear Dr. Roep,

Thank you for submitting your manuscript to PLOS ONE. After careful consideration, we feel that it has merit but does not fully meet PLOS ONE’s publication criteria as it currently stands. Therefore, we invite you to submit a revised version of the manuscript that addresses the points raised during the review process.

We look forward to receiving your revised manuscript.

Kind regards,

Matthias G von Herrath, MD PhD

Academic Editor

PLOS ONE

Journal Requirements:

Additional Editor Comments:

Dear investigators, Both reviewers raised foremost the point that the cohort size you are studying does not fully support your conclusions. this needs to be addressed, either by toning things down substantially or by expanding the studies with a validation cohort. Please let us know, what you decide to do. matthias von Herrath, MD

Reviewers' comments:

Reviewer's Responses to Questions

**Comments to the Author**

1. Is the manuscript technically sound, and do the data support the conclusions?

Reviewer #1: Partly

Reviewer #2: No

2. Has the statistical analysis been performed appropriately and rigorously? 

Reviewer #1: Yes

Reviewer #2: Yes

3. Have the authors made all data underlying the findings in their manuscript fully available?

Reviewer #1: Yes

Reviewer #2: Yes

4. Is the manuscript presented in an intelligible fashion and written in standard English?

Reviewer #1: Yes

Reviewer #2: Yes

5. Review Comments to the Author

Reviewer #1: Roep et al. state that they hypothesize that ‘type 1 diabetes patients receiving the current standard of care (intensive insulin therapy) and carrying a copy of the protective INS variant have a further reduced risk of developing diabetes complications compared to patients only carrying the susceptible INS variant’. However, the results are more focused on having the INS variant or not rather than intensive treatment with the variant vs. non-intensive therapy with the protective variant. This is a bit confusing.

The authors should also provided a brief summary of what is meant by primary & secondary prevention from DCCT/EDIC as not all readers may be familiar with the differences, including the differences in the mean time from diagnosis & the presence of mild/moderate retinopathy at baseline in the ‘secondary prevention’ cohort.

The numbers used in this current analyses are quite small for making strong interpretations of associations—will the authors further evaluate this in a separate cohort to assess the impact of the protective variant on progression to complications (or worsening of complications)? I appreciate the challenges with the expected sample limitations when discussing precision approaches to treating T1D or associated complications but the authors should provide more discussion regarding the limitations and how might the results be strengthened.

Reviewer #2: The authors Tienhoven and co-workers evaluated insulin gene polymorphism in 1,363 T1D patients. They then determined if there was an association between the protective INS variant and diabetic complications viz., proliferative diabetic retinopathy (PDR) and diabetic kidney disease (DKD). The authors concluded that the protective INS gene variant imparted some degree of protection against PDR but not DKD in intensively treated patient sub-population. In the conventionally treated arm, there was no difference in the rate of diabetic complication.

Comments:

1. There seems to be discrepancy in the number of patients (carrying the susceptible INS variant, in the primary prevention group) developing PDR or DKD in intensively treated arm i.e., in the text the authors write,”11/258 lacking the protective INS variant developed PDR (n=5), DKD (n=3), or both (n=3)”. However, in the figure legend (of Fig 1), the authors write,”however, PDR and DKD was observed in those carrying the susceptible INS variant (8/258 and 6/258 respectively)”.

2. In the secondary intervention group, intensive treatment in patients with protective INS variant has significantly less incidence of PDR while that of DKD did not change. Intensive insulin therapy in T2D is known to impact retinal vasculature. Is it plausible then that similar phenomenon might be occurring in T1D patients? Besides, is it possible there are other unknown epigenetic factors contributing to the observed effect of PDR.

3. In the conventionally treated patient group there was no difference in the frequency of developing diabetic complications in the primary prevention group regardless of the INS variant. This suggests that intensive treatment regimen is necessary for the suggested “protective” impact of INS variant on PDR.

Overall, while the authors put forth a potentially interesting hypothesis, the study subgroup is under-powered to enable such firm conclusion. It would immensely strengthen this interesting observation/hypothesis if the authors can replicate their findings in another cohort.

6. PLOS authors have the option to publish the peer review history of their article (what does this mean?). If published, this will include your full peer review and any attached files.

Reviewer #1: No

Reviewer #2: No

---

## [Author Response · Author response to Decision Letter 0]

15 Dec 2022

Editor Comments:

Dear investigators, 

Both reviewers raised foremost the point that the cohort size you are studying does not fully support your conclusions. this needs to be addressed, either by toning things down substantially or by expanding the studies with a validation cohort. Please let us know, what you decide to do. matthias von Herrath, MD

We are grateful for the thoughtful and constructive responses from the Reviewers and Associate Editor. We appreciate the value of validation cohorts and had thoroughly considered our options. Unfortunately, this proved to be challenging for several reasons:

1. The DCCT/EDIC cohort is the largest of its kind with the longest follow-up.

2. Other T1D cohorts do not (yet) feature long term diabetic complications (i.e., T1DGC, TrialNet, TRGIR, FR1DA, INNODIA and TEDDY). One cohort (GENIE) that does carry information on diabetic complications defined DKD differently (persistent proteinuria >05 g/24 hours), did not measure PDR, and did not include any data on the history of T1D treatment (see point 3 that follows). Thus, we were not able to directly compare our findings with data from that cohort. 

3. A further complication involved the relationship with glycemic control that we report: it matters whether T1D patients receive intensive insulin therapy and guidance, which happened to be the incentive of the DCCT cohort.

As we are aware that obtaining corroborating results in other cohorts could strengthen our results and interpretation, we were delighted to note that a sufficiently large pediatric cohort (therefore lacking data on complications) showed proof in the very same direction that carrying a copy of the protective INS variant has favorable outcome (in their case on glycemic control, preserved beta-cell function and lack of insulin autoimmunity) that are known to significantly reduce or even prevent diabetic complication. This independent observation from Nielsen and colleagues gave us indirect support for our observations in DCCT. We added a sentence in our discussion pointing to the desire for future validation (page 9-10, lines 158-165), in appreciation of this notion:

‘Although the DCCT/EDIC cohort is the largest T1D complications cohort with the longest follow-up time, a limitation of this study is the relatively small sample size. Alternative T1D cohorts were considered to replicate our results but these cohorts had even smaller numbers of study subjects, diabetic complications were not (yet) recorded or defined differently, or the relevant SNP typing was missing. Fortunately, the Hvidøre pediatric cohort showed proof in the same direction that carrying a copy of the protective INS variant has favorable outcome: in their case on glycemic control and preserved beta cell function [Nielsen, Diabetologia, 2006], both known to significantly reduce or even prevent diabetic complications, and thus indirectly yet independently supporting our observations in the DCCT cohort.’

We further tone our report down by stating (Discussion, page 9, lines 144-153):

‘The relationship between glycemic control and the association between INS variation and diabetic complication remains unclear. In the DCCT cohort, HbA1c levels were much better in the intensive treatment arm compared to the conventional treatment arm but did not differ between patients with and without the protective INS variant within the intensive treatment arm. Yet, in the Hvidøre cohort an association between superior glycemic control and the protective INS variant was observed in children in the first year after their T1D diagnosis [Nielsen, Diabetologia, 2006]. We speculate that the impact of the protective INS variant on complication probability involves epigenetic modifications early after disease onset (‘metabolic memory’) and may change with improved glycemic control later in disease progression [Chen, Nat Metabolism, 2020]. Future and independent validation of our findings in other large cohorts therefore remains warranted.’

 

Reviewers' comments:

Reviewer's Responses to Questions

Comments to the Author

1. Is the manuscript technically sound, and do the data support the conclusions?

Reviewer #1: Partly

Reviewer #2: No

2. Has the statistical analysis been performed appropriately and rigorously?

Reviewer #1: Yes

Reviewer #2: Yes

3. Have the authors made all data underlying the findings in their manuscript fully available?

Reviewer #1: Yes

Reviewer #2: Yes

4. Is the manuscript presented in an intelligible fashion and written in standard English?

Reviewer #1: Yes

Reviewer #2: Yes

5. Review Comments to the Author

 

Reviewer #1: Roep et al. state that they hypothesize that ‘type 1 diabetes patients receiving the current standard of care (intensive insulin therapy) and carrying a copy of the protective INS variant have a further reduced risk of developing diabetes complications compared to patients only carrying the susceptible INS variant’. However, the results are more focused on having the INS variant or not rather than intensive treatment with the variant vs. non-intensive therapy with the protective variant. This is a bit confusing.

Thank you for this comment. We apologize for this confusion. The DCCT/EDIC trial contains two treatment arms: one where patients received intensive treatment and the other with patients receiving the treatment that was conventional at the time of the trial. We followed this stratification for our analysis and focused on the intensive treatment arm, since this is the standard of clinical care and therefore most relevant to patients nowadays. The INS variants were actually tested in both treatment arms to show that the rate of glycemic control (which by definition is better in the ‘intensive treatment’ arm) matters. In the intensive treatment arm DKD and PDR were completely absent in patients with the protective INS variant in the primary prevention group and PDR was significantly less frequent in patients with the protective INS variant in the secondary intervention group. In the conventional treatment arm, we report no difference in DKD or PDR occurrence, except that no DKD was observed in patients homozygous for the protective INS variant. For this reason, we stated that carrying a protective INS variant only reduces the risk of complications in patients that received the current standard of care (which is intensive insulin treatment). We hope this clears the confusion that our wording had caused and clarified the methods of our manuscript accordingly (page 4, lines 73-79):

‘The DCCT/EDIC trial contained two treatment arms (intensive insulin therapy vs conventional therapy), and two cohorts (primary prevention and secondary intervention) within each arm (four groups in total). The association between INS variation and complication probability was analyzed for each group separately. The patients in the primary prevention cohort had T1D for 1-5 years without evidence of retinopathy and albumin secretion rates of <40mg/24h. The patients in the secondary intervention cohort had T1D for 1-15 years and at least one microaneurysm in either eye and albumin secretion rates no greater than 200mg/24h.’

In addition, to further emphasize the difference between the intensive and conventional treatment arm, we revised the presentation of the HbA1c data in Table 1 to stratify the results by both therapy arms and insulin variant (not just INS variant alone). Please see the revised table below. We also clarified the relationship between glycemic control and the association between INS variation and diabetic complications in the discussion of our revised manuscript (page 9, lines 144-153):

‘The relationship between glycemic control and the association between INS variation and diabetic complication remains unclear. In the DCCT cohort, HbA1c levels were much better in the intensive treatment arm compared to the conventional treatment arm but did not differ between patients with and without the protective INS variant within the intensive treatment arm. Yet, in the Hvidøre cohort an association between superior glycemic control and the protective INS variant was observed in children in the first year after their T1D diagnosis [Nielsen, Diabetologia 2006]. We speculate that the impact of the protective INS variant on complication probability involves epigenetic modifications early after disease onset and may change with improved glycemic control later in disease progression [Chen, Nat Metabolism, 2020]. Future and independent validation of our findings in other large cohorts therefore remains warranted.’

 

Characteristic DCCT/EDIC Patients by Insulin Variant

 Protective Susceptible

 n Value* n Value*

Age at T1D Onset (years)† 275 20.8 (±8.0) 1088 21.3 (±8.1)

Sex 

 Female

 Male 

133

142 

48%

52% 

509

579 

47%

53%

Race 

 White

 Black

 Hispanic

 Asian/Pacific Islander

 American Indian/Alaska Native 

266

7

2

0

0 

97%

2%

1%

0%

0% 

1047

22

12

6

1 

96.4%

2%

1%

0.5%

0.1%

BMI (kg/m2)

 DCCT Start

 DCCT End (Y9)

 EDIC Start

 EDIC Last Available (Y8) 

275

253

254

242 

23.2 (±2.7)

25.4 (±3.4)

25.6 (±3.5)

27.6 (±4.1) 

1088

1003

1002

971 

23.5 (±2.8)

25.8 (±3.8)

25.8 (±3.7)

27.5 (±4.5)

HbA1c (%)††

 Intensive therapy arm

 DCCT Start

 DCCT End (Y9)

 Conventional treatment arm 

 DCCT Start

 DCCT End (Y9) 

150

149

125

124 

8.9 (±1.7)

7.5 (±1.1)

8.9 (±1.8)

9.2 (±1.6) 

518

515

570

569 

8.9 (±1.5)

7.4 (±1.1)

8.9 (±1.6)

9.1 (±1.5)

Post disease diagnosis follow up period (years)

 Intensive arm

 Primary prevention group

 Secondary intervention group

 Conventional treatment arm

 Primary prevention group

 Secondary intervention group 

67

83

66

59 

20.4 (±1.6)

26.1 (±3.9)

20.2 (±2.0)

25.6 (±4.6) 

258

260

292

278 

20.4 (±1.8)

26.3 (±4.1)

19.8 (±2.7)

25.1 (±4.5)

 

The authors should also provided a brief summary of what is meant by primary & secondary prevention from DCCT/EDIC as not all readers may be familiar with the differences, including the differences in the mean time from diagnosis & the presence of mild/moderate retinopathy at baseline in the ‘secondary prevention’ cohort.

Thank you for this comment. We added a brief statement regarding the differences between primary prevention and secondary intervention in the methods section of our revised manuscript (page 4, lines 73-79):

‘The DCCT/EDIC trial contained two treatment arms (intensive insulin therapy vs conventional therapy), and two cohorts (primary prevention and secondary intervention) within each arm (four groups in total). The association between INS variation and complication probability was analyzed for each group separately. The patients in the primary prevention cohort had T1D for 1-5 years without evidence of retinopathy and albumin secretion rates of <40mg/24h. The patients in the secondary intervention cohort had T1D for 1-15 years and at least one microaneurysm in either eye and albumin secretion rates no greater than 200mg/24h.’

The numbers used in this current analyses are quite small for making strong interpretations of associations—will the authors further evaluate this in a separate cohort to assess the impact of the protective variant on progression to complications (or worsening of complications)? I appreciate the challenges with the expected sample limitations when discussing precision approaches to treating T1D or associated complications but the authors should provide more discussion regarding the limitations and how might the results be strengthened.

Thank you for this comment. We appreciate the value of validation cohorts and had thoroughly considered our options. Unfortunately, this proved to be challenging for several reasons:

1. The DCCT/EDIC cohort is the largest of its kind with the longest follow-up.

2. Other T1D cohorts do not (yet) feature long term diabetic complications (i.e., T1DGC, TrialNet, TRGIR, FR1DA, INNODIA and TEDDY). One cohort (GENIE) that does carry information on diabetic complications defined DKD differently (persistent proteinuria >05 g/24 hours), did not measure PDR, and did not include any data on the history of T1D treatment (see point 3 that follows). Thus, we were not able to directly compare our findings with data from that cohort. 

3. A further complication involved the relationship with glycemic control that we report: it matters whether T1D patients receive intensive insulin therapy and guidance, which happened to be the incentive of the DCCT cohort.

As we are equally aware that finding corroborating results in other cohorts could strengthen our results and interpretation, we were delighted to note that a sufficiently large pediatric cohort (therefore lacking data on complications) showed proof in the very same direction that carrying a copy of the protective INS variant has favorable outcome (in their case on glycemic control, preserved beta-cell function and lack of insulin autoimmunity) that are known to significantly reduce or even prevent diabetic complication. This independent observation from Nielsen and colleagues gave us indirect support for our observations in DCCT. We added a sentence in our discussion pointing to the desire for future validation (page 9-10, lines 158-165), in appreciation of this notion:

‘Although the DCCT/EDIC cohort is the largest T1D complications cohort with the longest follow-up time, a limitation of this study is the relatively small sample size. Alternative T1D cohorts were considered to replicate our results but these cohorts had even smaller numbers of study subjects, diabetic complications were not (yet) recorded or defined differently or the relevant SNP typing was missing. Fortunately, the Hvidøre pediatric cohort showed proof in the same direction that carrying a copy of the protective INS variant has favorable outcome: in their case on glycemic control and preserved beta cell function [Nielsen, Diabetologia, 2006], both known to significantly reduce or even prevent diabetic complications, and thus indirectly yet independently supporting our observations in the DCCT cohort.’

We further tone our report down by stating (Discussion, page 9, lines 144-153):

‘The relationship between glycemic control and the association between INS variation and diabetic complication remains unclear. In the DCCT cohort, HbA1c levels were much better in the intensive treatment arm compared to the conventional treatment arm but did not differ between patients with and without the protective INS variant within the intensive treatment arm. Yet, in the Hvidøre cohort an association between superior glycemic control and the protective INS variant was observed in children in the first year after their T1D diagnosis [Nielsen, Diabetologia, 2006]. We speculate that the impact of the protective INS variant on complication probability involves epigenetic modifications early after disease onset (‘metabolic memory’) and may change with improved glycemic control later in disease progression [Chen, Nat Metabolism, 2020]. Future and independent validation of our findings in other large cohorts therefore remains warranted.’

 

Reviewer #2: The authors Tienhoven and co-workers evaluated insulin gene polymorphism in 1,363 T1D patients. They then determined if there was an association between the protective INS variant and diabetic complications viz., proliferative diabetic retinopathy (PDR) and diabetic kidney disease (DKD). The authors concluded that the protective INS gene variant imparted some degree of protection against PDR but not DKD in intensively treated patient sub-population. In the conventionally treated arm, there was no difference in the rate of diabetic complication.

Comments:

1. There seems to be discrepancy in the number of patients (carrying the susceptible INS variant, in the primary prevention group) developing PDR or DKD in intensively treated arm i.e., in the text the authors write,”11/258 lacking the protective INS variant developed PDR (n=5), DKD (n=3), or both (n=3)”. However, in the figure legend (of Fig 1), the authors write,”however, PDR and DKD was observed in those carrying the susceptible INS variant (8/258 and 6/258 respectively)”.

Thank you for this comment. We appreciate that the reporting of these numbers was confusing, though correct. In the primary prevention group 11 patients developed complications, of which some developed both: 5 patients developed PDR only, 3 patients developed DKD only, and 3 patients developed both PDR and DKD. Therefore, the count of PDR occurrences was (5 alone + 3 both=)8 and the count of DKD occurrences was (3 alone + 3 both=)6. This has now been specified in our revised manuscript (page 7, lines 102-104): 

‘However, 11 out of 258 patients (4.3%) lacking the protective INS variant developed complications (PDR only n=5, DKD only n=3, or both PDR and DKD n=3) in 20.4 ± 1.8 years of follow-up after diagnosis (Fig 1).’

and (page 8, lines 121-122): 

‘however, PDR and DKD occurred in those carrying the susceptible INS variant (8 of 258 and 6 of 258, respectively).’

2. In the secondary intervention group, intensive treatment in patients with protective INS variant has significantly less incidence of PDR while that of DKD did not change. Intensive insulin therapy in T2D is known to impact retinal vasculature. Is it plausible then that similar phenomenon might be occurring in T1D patients? Besides, is it possible there are other unknown epigenetic factors contributing to the observed effect of PDR.

Thank you for raising these interesting thoughts. It is certainly plausible that similar features may apply in T2D. We assume that T2D patients carrying the INS variant protecting from T1D have better glycemic control than those carrying the susceptible INS variant, but this has not yet been investigated. While beyond the scope of our current report, epigenetic factors indeed contribute to (reduced) risk for PDR, as we recently reported that DNA methylation mediated development of HbA1c-associated complications in type 1 diabetes in the DCCT cohort (Nature Metab, 2020). It is not yet known whether these involve the INS region. We added a sentence speculating on this in the Discussion (page 9, lines 149-152):

‘We speculate that the impact of the protective INS variant on complication probability involves epigenetic modifications early after disease onset and may change with improved glycemic control later in disease progression [ref Chen, Nat Metabolism, 2020].’

3. In the conventionally treated patient group there was no difference in the frequency of developing diabetic complications in the primary prevention group regardless of the INS variant. This suggests that intensive treatment regimen is necessary for the suggested “protective” impact of INS variant on PDR.

Overall, while the authors put forth a potentially interesting hypothesis, the study subgroup is under-powered to enable such firm conclusion. It would immensely strengthen this interesting observation/hypothesis if the authors can replicate their findings in another cohort.

Thank you for this comment. We appreciate the value of validation cohorts and had thoroughly considered our options. Unfortunately, this proved to be challenging for several reasons:

1. The DCCT/EDIC cohort is the largest of its kind with the longest follow-up.

2. Other T1D cohorts do not (yet) feature long term diabetic complications (i.e., T1DGC, TrialNet, TRGIR, FR1DA, INNODIA and TEDDY). One cohort (GENIE) that does carry information on diabetic complications defined DKD differently (persistent proteinuria >05 g/24 hours), did not measure PDR, and did not include any data on the history of T1D treatment (see point 3 that follows). Thus, we were not able to directly compare our findings with data from that cohort. 

3. A further complication involved the relationship with glycemic control that we report: it matters whether T1D patients receive intensive insulin therapy and guidance, which happened to be the incentive of the DCCT cohort.

As we are equally aware that finding corroborating results in other cohorts could strengthen our results and interpretation, we were delighted to note that a sufficiently large pediatric cohort (therefore lacking data on complications) showed proof in the very same direction that carrying a copy of the protective INS variant has favorable outcome (in their case on glycemic control, preserved beta-cell function and lack of insulin autoimmunity) that are known to significantly reduce or even prevent diabetic complication. This independent observation from Nielsen and colleagues gave us indirect support for our observations in DCCT. We added a sentence in our discussion pointing to the desire for future validation (page 9-10, lines 158-165), in appreciation of this notion:

‘Although the DCCT/EDIC cohort is the largest T1D complications cohort with the longest follow-up time, a limitation of this study is the relatively small sample size. Alternative T1D cohorts were considered to replicate our results but these cohorts had even smaller numbers of study subjects, diabetic complications were not (yet) recorded or defined differently or the relevant SNP typing was missing. Fortunately, the Hvidøre pediatric cohort showed proof in the same direction that carrying a copy of the protective INS variant has favorable outcome: in their case on glycemic control and preserved beta cell function [Nielsen, Diabetologia, 2006], both known to significantly reduce or even prevent diabetic complications, and thus indirectly yet independently supporting our observations in the DCCT cohort.’

We further tone our report down by stating (Discussion, page 9, lines 144-153):

‘The relationship between glycemic control and the association between INS variation and diabetic complication remains unclear. In the DCCT cohort, HbA1c levels were much better in the intensive treatment arm compared to the conventional treatment arm but did not differ between patients with and without the protective INS variant within the intensive treatment arm. Yet, in the Hvidøre cohort an association between superior glycemic control and the protective INS variant was observed in children in the first year after their T1D diagnosis [Nielsen, Diabetologia, 2006]. We speculate that the impact of the protective INS variant on complication probability involves epigenetic modifications early after disease onset (‘metabolic memory’) and may change with improved glycemic control later in disease progression [Chen, Nat Metabolism, 2020]. Future and independent validation of our findings in other large cohorts therefore remains warranted.’

---

## [Decision Letter · Decision Letter 1]

11 Jan 2023

Low risk for diabetic complications in type 1 diabetes patients carrying a protective insulin gene variant

PONE-D-22-27736R1

Dear Dr. Roep,

We’re pleased to inform you that your manuscript has been judged scientifically suitable for publication and will be formally accepted for publication once it meets all outstanding technical requirements.

Kind regards,

Matthias G von Herrath, MD PhD

Academic Editor

PLOS ONE

Additional Editor Comments (optional):

Reviewers' comments:

Reviewer's Responses to Questions

**Comments to the Author**

1. If the authors have adequately addressed your comments raised in a previous round of review and you feel that this manuscript is now acceptable for publication, you may indicate that here to bypass the “Comments to the Author” section, enter your conflict of interest statement in the “Confidential to Editor” section, and submit your "Accept" recommendation.

Reviewer #2: All comments have been addressed

2. Is the manuscript technically sound, and do the data support the conclusions?

Reviewer #2: Yes

3. Has the statistical analysis been performed appropriately and rigorously? 

Reviewer #2: Yes

4. Have the authors made all data underlying the findings in their manuscript fully available?

Reviewer #2: Yes

5. Is the manuscript presented in an intelligible fashion and written in standard English?

Reviewer #2: Yes

6. Review Comments to the Author

Reviewer #2: Thank you for addressing the concerns ! I do hope that you will attempt to validate these findings in an independent cohort - as and when that might be available.

7. PLOS authors have the option to publish the peer review history of their article (what does this mean?). If published, this will include your full peer review and any attached files.

Reviewer #2: No

---

## [Editor Report · Acceptance letter]

16 Jan 2023

PONE-D-22-27736R1 

Low risk for diabetic complications in type 1 diabetes patients carrying a protective insulin gene variant 

Dear Dr. Roep:

I'm pleased to inform you that your manuscript has been deemed suitable for publication in PLOS ONE. Congratulations! Your manuscript is now with our production department. 

Kind regards, 

on behalf of

Prof. Matthias G von Herrath 

Academic Editor

PLOS ONE